# Shiftless Is a Novel Member of the Ribosome Stress Surveillance Machinery That Has Evolved to Play a Role in Innate Immunity and Cancer Surveillance

**DOI:** 10.3390/v15122296

**Published:** 2023-11-23

**Authors:** Jamie A. Kelly, Jonathan D. Dinman

**Affiliations:** Department of Cell Biology and Molecular Genetics, University of Maryland, College Park, MD 20742, USA; twz3@cdc.gov

**Keywords:** ribosome, frameshifting, translation, fidelity, shiftless, RQC, virus, innate immunity, stress, surveillance

## Abstract

A longstanding paradox in molecular biology has centered on the question of how very long proteins are synthesized, despite numerous measurements indicating that ribosomes spontaneously shift reading frame at rates that should preclude their ability completely translate their mRNAs. Shiftless (SFL; C19orf66) was originally identified as an interferon responsive gene encoding an antiviral protein, indicating that it is part of the innate immune response. This activity is due to its ability to bind ribosomes that have been programmed by viral sequence elements to shift reading frame. Curiously, Shiftless is constitutively expressed at low levels in mammalian cells. This study examines the effects of altering Shiftless homeostasis, revealing how it may be used by higher eukaryotes to identify and remove spontaneously frameshifted ribosomes, resolving the apparent limitation on protein length. Data also indicate that Shiftless plays a novel role in the ribosome-associated quality control program. A model is proposed wherein SFL recognizes and arrests frameshifted ribosomes, and depending on SFL protein concentrations, either leads to removal of frameshifted ribosomes while leaving mRNAs intact, or to mRNA degradation. We propose that SFL be added to the growing pantheon of proteins involved in surveilling translational fidelity and controlling gene expression in higher eukaryotes.

## 1. Introduction

Protein synthesis is the most energy intensive of all cellular processes [1,2]. Accordingly, protein synthesis is controlled and monitored at many levels in order to maximize cellular energy utilization by rapidly identifying and responding to translational errors [3]. One of the core requirements for efficient protein synthesis is that ribosomes faithfully maintain translational reading frame. As evidence of this, the intrinsic rate of spontaneous ribosomal frameshifting (sRF) has been measured to be very low, on the order of one event per 5000 codons or ~2 × 10^−4^ [4,5,6,7,8]. Although this should not present a problem for simple organisms whose coding sequences fall below this limit, it poses a conundrum for higher eukaryotes which utilize larger proteins. For example, at the rate of 2 × 10^−4^ frameshifts/codon, the probability that any single ribosome would be able to fully translate an mRNA of 3500 codons without shifting reading frame is approximately 50% (Appendix A). Decreasing the rate of random frameshifting to 10^−4^ only raises this threshold to ~7000 codons and doubling the rate to 4 × 10^−4^ lowers this threshold to ~1700 codons. Given that the human genome encodes at least 23 proteins greater than 5000 codons, some of which also have a broad tissue distribution [9], the energetic costs incurred by their translation would be prohibitive. Taking this analysis further, it should be impossible to fully synthesize very large proteins such as Titan, whose mRNA encodes 38,138 amino acid residues [10]. Additionally, accumulation of truncated proteins resulting from spontaneous frameshifts can have trans-dominant negative and/or deleterious gain-of-function effects on cell growth and homeostasis [11]. In short, how mRNAs encoding very long proteins can be efficiently synthesized is a longstanding unsolved paradox of molecular biology.

Despite strong limitations on spontaneous ribosomal frameshifting, many RNA viruses evade this by programming ribosomes to shift into alternate reading frames, often at frequencies of three orders of magnitude greater than the background rate, through a process called Programmed Ribosomal Frameshifting (PRF) [12,13,14,15]. It is generally accepted that RNA viruses use PRF to maximize their gene expression programs without altering their genomes [16]. PRF has also been documented to occur on mRNAs of cellular origin [17,18,19], although this field remains controversial due to issues recently raised regarding dual luciferase reporter systems that had been in wide use for approximately two decades [20]. Regardless, PRF is directed to occur at specific locations along mRNAs, typically by a combination of specific ‘slippery’ sequences and structured RNA elements [12,21]. Experimentally, altering viral PRF frequencies has negative consequences on viral particle biogenesis, thus rendering PRF a focus for development of antiviral therapeutics [22,23,24]. In the evolutionary battle between viruses and their host cells, this reliance of viruses for high and precise rates of PRF has been targeted by higher eukaryotes in the form of a protein called Shiftless (a.k.a. C19orf66). Shiftless (SFL) recognizes and binds to ribosomes that are in the process of shifting reading frame, arresting them at the PRF signals [25]. While much remains unknown about how SFL works, it appears to lock frameshifting ribosomes in place on the mRNA and recruits eRF3 (and presumably its partner eRF1) to terminate translation by frameshifted ribosomes. Ongoing research efforts are revealing that SFL overexpression inhibits the replication of a growing number of viruses [25,26,27]. Not surprisingly, SFL transcription is upregulated by interferons, i.e., SFL is an interferon responsive gene (IRG), identifying it as a participant in the innate immune response [27].

Curiously, an immunoblot analysis indicated that SFL appears to be constitutively expressed in mammalian cells, albeit at much lower levels than during viral infection [28]. The current study addresses the reason for constitutive SFL expression. In so doing, it illuminates how SFL may be used by higher eukaryotes to overcome the apparent limitation on protein length. Genetic evidence is also presented connecting SFL with the molecular apparatus involved in translational surveillance known as ribosome stress surveillance [29]. A model is proposed wherein SFL recognizes and arrests frameshifted ribosomes, and depending on SFL protein concentrations, either leads to removal of frameshifted ribosomes while leaving mRNAs intact, or to mRNA degradation. As such, we propose that SFL be added to the growing pantheon of proteins involved in surveilling translational fidelity and controlling gene expression in higher eukaryotes.

## 2. Materials and Methods

### 2.1. Cell Culture and Plasmid Transfections

Human embryonic kidney (HEK293T/17) (CRL-11268), HeLa (CCL-2), and U87 MG (HTB-14) cells were purchased from the American Type Culture Collection (Manassas, VA, USA). HEK 293T cells were maintained in Dulbecco’s modified Eagle’s medium (Corning Life Sciences 10-013-CV Durham, NC, USA) supplemented with 10% fetal bovine serum (Life Technologies 26140079 Carlsbad, CA, USA) 1% GlutaMAX (35050061), 1% nonessential amino acids (Gibco 11140050), 1% HEPES buffer (Cytiva Life Sciences SH30237.01) and 1x Penicillin/Streptomycin (Gibco 15140122). HeLa and U87 MG cells were maintained in Dulbecco’s modified Eagle’s medium supplemented with 10% fetal bovine serum, 1% GlutaMAX, and 1x Penicillin/Streptomycin. Both cell lines were incubated at 37 °C in 5% CO_2_. The cells were transfected with a total of 500 ng (dual luciferase assay) or 1 µg (bifluorescence and qRT PCR assays) plasmid DNA 24 h after seeding using Lipofectamine 3000 (Invitrogen L3000015) per the manufacturer’s protocol.

### 2.2. shRNA Knockdowns

A panel of short hairpin shRNA plasmids targeting SFL and commercially validated shRNAs targeting Pelota, SMG1, ASCC3, Hbs1L, and ZNF598 were purchased from Millipore Sigma. shRNA targets are listed in Table 1. Five different SFL shRNAs (SFL shA–E) were assayed and knockdowns were conducted by transfecting 250 ng (dual luciferase assay) or 500 ng (bifluorescence assay) shRNA plasmid DNA into HEK293T cells 24 h after seeding using Lipofectamine 3000 per the manufacturer’s protocol. RT-PCR analysis identified two shRNAs, shRNA A and shRNA B, that promoted the strongest knockdown of SFL, and quantitatively validated by qRT PCR (Appendix A).

### 2.3. Generation of CRISPR Knockout Cell Lines

An SFL^-/-^ HEK293T cell line was generated using CRSIPR-Cas9 as described in Ran et al. [30] HEK923T cells were transfected with spCas9(BB)-2A-Puro plasmid containing a gRNA specific to C19orf66 (5′ CGTGTATCCAACACGGATCC 3′) designed to result in a 1 base deletion in the ORF using Lipofectamine3000 per manufacturer’s protocol. Cells were selected for the presence of Cas9 by incubating with 1.0 µg/mL puromycin. Clonal cell lines were obtained by seeding cells at low density in a 10 cm tissue culture dish and then selecting well-isolated colonies for expansion. Clonal lines were screened for mutations by PCR-amplifying C19orf66 from genomic DNA and assessing for mutations using the Surveyor mutation detection kit (Integrated DNA Technologies 706020) per manufacturer’s protocol. Positive hits from the Surveyor screen were sequence-verified (Appendix A) and knockout of SFL was validated via qRT PCR (Appendix A).

### 2.4. Growth Curve of SFL^-/-^ HEK293T Cells

HEK WT and SFL^-/-^ cells were seeded at a density of 10^4^ cells per well of a 12-well plate. Cells were trypsinized and counted at 24 h intervals for a total of 168 h. Cell doubling times were determined using the equation:$$DT=\frac{T*\ln\left(2\right)}{\ln\left(\frac{X_{e}}{X_{b}}\right)}$$
where *T* = time in hours, *X_e_* = endpoint cell count, *X_b_* = beginning cell count.

These data are shown in Appendix A.

### 2.5. Preparation of Reporter Plasmids

Dual luciferase reporters containing -1 PRF signals of SARS-CoV, SARS-CoV-2, and human CCR5 listed in Table 2 were made by digesting pJD2257 with *Sal* I and *Bam* HI, gel-purifying digest products, and ligating a DNA oligonucleotide insert (IDT) containing the -1 PRF signal of interest into the plasmid using T4 DNA ligase (NEB). pJD2257 is derived from pSGDLuc [20] into which *Sal* I and *Bam* HI sites were inserted so as to preclude possible distortions of luciferase reporter readouts due to insertion of sequences of viral or cellular origin. The spontaneous -1 frameshift (-1 sFS) reporter plasmid was made using site-directed mutagenesis of pJD2257. Site directed mutagenesis primers (Table 3) were synthesized and purified by IDT.

A bifluorescent reporter was initially constructed by swapping AcGFP and mCherry coding sequences into pJD2257, replacing the dual luciferase cassettes to construct pJD2262. A panel of bifluorescent reporters containing human and viral translational control element inserts (Table 2) were made by digesting pJD2261 with *Sal* I and *Bam* HI, gel-purifying digest products, and ligating a DNA oligonucleotide insert (IDT) containing sequences of interest into the plasmid using T4 DNA ligase (NEB). Products were transformed into DH5α Escherichia coli cells (NEB) and spread onto LB agar plates containing 50 mg/mL carbenicillin. Positive clones were verified by DNA sequencing (Genewiz).

### 2.6. Dual Luciferase Assays of -1 PRF

The frameshifting efficiency of luciferase reporter plasmids in cultured cells was assayed as previously described using a dual luciferase reporter assay system kit (Promega Madison, WI, USA) [31,32]. Then, 24 h after transfection, cells were washed with 1x PBS then lysed with 1x passive lysis buffer (E194A, Promega). Reporter activity was calculated by measuring the luminescence of firefly or Renilla luciferase in 50 µL of cell lysate. Assays were conducted in triplicate in 96-well plates and quantified using a GloMax microplate luminometer (Promega). Percent frameshift was calculated by averaging the three Firefly or Renilla luciferase technical replicate reads per sample then forming a ratio of firefly to Renilla luminescence per sample. Each sample ratio was compared to a 0-frame control set to 100%. At least three biological replicates with three technical replicates each were assayed for each sample. Statistical analyses were conducted using one-way analysis of variance using Prism 9 software (GraphPad).

### 2.7. Bifluorescence Assays of -1 PRF

Frameshift efficiency of bifluorescent reporters in cultured cells were assayed as described previously [33]. HEK293T or HeLa cells were seeded at density of 10^5^ cells per well of a 12-well plate in appropriate growth media. After a 24 h incubation, cells were transfected with 500 ng bifluorescence reporter plasmid and 500 ng of either a SFL overexpression plasmid, shRNA plasmid targeting SFL, or shRNA scramble non-targeting control. Additional media was added to cells 24 h post transfection and incubated for an additional 24 h. Cells were collected by scraping into Dulbecco’s phosphate-buffered saline (Corning 21-031-CV), pelleted by centrifugation then lysed in 150 uL Triton lysis buffer (1% Triton X-100, 150 mM NaCl, 50 mM Tris pH8, 1x Halt protease inhibitor cocktail (Thermo Scientific Waltham, MA, USA)). Cell lysates were clarified by centrifugation and assayed in a clear-bottom black-walled 96-well plate (Grenier Bio-One Atlanta, GA, USA). Fluorescence was quantified using a GloMax microplate luminometer (Promega Madison, WI, USA) with the “green” optical kit (Excitation 525 nm, Emission 580–640 nm) for mCherry and the “blue” optical kit (excitation 490 nm, emission 510–570 nm) for AcGFP. Reporter activity and -1 PRF efficiency was corrected for AcGFP bleed over into the mCherry channel by subtracting background fluorescence from mock transfected cells. Reporter activity was calculated by measuring the fluorescence of 150 µL clarified cell lysate and subtracting background fluorescence from mock transfected cells. -1 PRF efficiency was calculated as previously described [33]. Statistical analyses were conducted using one-way analysis of variance using Prism 9 software (GraphPad).

### 2.8. qRT PCR and RT PCR Methods

To quantitatively measure mRNA abundances (qRT-PCR), total RNA was extracted from cells grown to 70–80% confluency using the Total RNA miniprep kit (NEB T2010S) according to the manufacturer’s protocol. cDNA synthesis was performed using the iScript gDNA Clear cDNA synthesis kit (BioRad 1725034) according to the manufacturer’s protocol using 500 ng total RNA. qPCR was performed with 100 ng of total cDNA, 250 nM primers in a final volume of 10 µL using BioRad CFX96 and SSOAdvanced SYBR green master mix (BioRad 1725270). mRNA abundances for SFL overexpression and SFL^-/-^ conditions were normalized to WT HEK293T conditions. Primers for this assay are listed in Table 3. qPCR conditions were 4 min at 95 °C followed by 40 cycles at 95 °C for 5 s and 60 °C for 30 s followed by a melt curve from 65 °C to 95 °C. Experiments were repeated with at least three independent biological replicates.

To monitor the presence of specific mRNAs (RT-PCR), RNA was extracted from HEK293T, HeLa, and U87MG cells grown to 70–80% confluency and an overnight *E. coli* culture using the Total RNA miniprep kit (NEB T2010S) according to the manufacturer’s protocol. cDNA synthesis was performed using the iScript gDNA Clear cDNA synthesis kit (BioRad 1725034) according to the manufacturer’s protocol. RT PCR was performed using 1 µL of cDNA, and 500 nM primers in a final volume of 35 µL using Q5 Hotstart Mastermix (NEB M0494S). Primers for this assay are listed in Table 3. PCR conditions were: 98 °C for 30 s followed by 40 cycles at 98 °C for 10 s, 71 °C for 15 s and 72 °C for 20 s, then 72 °C final extension for 2 min.

To assess splicing of reporter plasmid transcripts, RNA was extracted from HEK293T cells transfected with first-generation dual luciferase (p2luci), second-generation dual luciferase (pSGD), or bifluorescent (BiFl) reporters containing 0-frame control, CCR5 -1 PRF, or HIV-1 -1 PRF inserts. cDNA synthesis was performed as stated above. RT PCR was performed using 1 uL of cDNA or 100 pg of reporter plasmid as an input control, and 500 nM primers in a final volume of 20 uL using DreamTaq Green PCR Mastermix (Thermofisher K1081). Primers for this assay are listed in Table 3. PCR conditions were 95 °C for 2 min followed by 40 cycles at 95 °C for 30 s, 64.4 °C (p2luci), 57 °C (pSGD) or 59.8 °C (BiFl) for 30 s, and 72 °C for 3 min, then 72 °C final extension for 5 min.

### 2.9. Bioinformatics Analysis of SFL Expression in Cancer Cells

The expression of SFL was analyzed in common cancer tissues and normal tissues deposited in the TGCA database using Genome Browser [34] (http://genome.ucsc.edu (accessed on 4 February 2022)). Average transcripts per million (TPM) for each normal and cancerous cell type were displayed on a heatmap using GraphPad Prism 9 software. Survival corresponding to SFL expression levels among patients with various cancers was analyzed using the UALCAN platform [35].

## 3. Results

### 3.1. SFL Is Constitutively Expressed in Human-Derived Cells

Previous studies demonstrated that SFL is constitutively expressed at low levels in human-derived cell lines and can be upregulated by the type I interferon response [25,28]. To confirm that it is indeed constitutively expressed, we used RT-PCR to probe for SFL expression in RNAs extracted from three human derived cell lines, HeLa, HEK293T, and U87 MG cells. RNA from *E. coli* was used as a negative control. Our results show that both SFL and the shorter splice isoform, SFLS [25], are constitutively expressed in the human-derived cell lines but not in bacteria (Appendix A). In order to manipulate SFL levels in the cell, we used an overexpression plasmid, or short hairpin RNA (shRNAs) targeting SFL. Overexpressing SFL resulted in a 1.8-fold increase in SFL mRNA levels (*p* = 0.005), while SFL-targeting shRNAs shA and shB reduced mRNA 3.25 (*p* = 0.01) and 3.42-fold (*p* = 0.0098), respectively (Appendix A).

### 3.2. First-Generation Dual Luciferase Reporters, Not the CCR5 -1 PRF Element, Promote Off-Target mRNA Splicing

The first generation of dual luciferase plasmids used to assess frameshift efficiency, P2luci, were subject to multiple issues including off-target mRNA splicing and inefficient cleavage of reporter proteins resulting in inaccurate values of frameshifting. Though subsequent reporter generations largely resolved these issues, questions arose regarding if some frameshift elements were subject to off-target mRNA splicing. In particular, the validity of the human -1 PRF signal in CCR5 came into question after one report suggested apparent frameshifting of this sequence was a product of splicing and not a ribosomal frameshift [36]. If some reported -1 PRF signals are the result of off-target mRNA splicing and not a ribosomal frameshift event due to properties of the reporter system used or those intrinsic to the -1 PRF element itself, it is imperative to identify true frameshift elements from splicing artifacts. To ascertain whether the CCR5 -1 PRF reporter promoted off target mRNA splicing in the context of the second-generation dual luciferase and bifluorescent reporters, mRNAs were extracted from cells expressing these reporters, and primers complementary to the 5′ end of the 5′ reporters, and 3′ ends of the 3′ reporters were used to amplify the full-length mRNAs (Appendix A). Controls included the same reporters alone or containing the HIV-1 -1 PRF signal. The PCR products were separated by gel electrophoresis. Plasmid controls revealed bands of the expected lengths, i.e., ~2500 nt for the dual luciferase reporters and ~1500 nt for the bifluorescent reporters. Small molecular weight products were observed in the first generation (p2luci) 0-frame control and CCR5 -1 PRF reporter mRNA samples, consistent with the propensity of this reporter to undergo off-target splicing events [20]. These PCR products were absent from the second-generation (pSGD) and bifluorescent (BiFl) reporter samples. Additional PCR products of ~1200 and ~800 bp were observed with all the BiFl samples. In addition, no “reporter crash”, i.e., 3–4 order of magnitude decreases in raw reporter protein activities indicative of off-target splicing, were observed in CCR5 samples as compared to HIV-1 samples from either the pSGD or BiFl data (Appendix A). These findings indicate that off-target splicing was due to sequences in the first-generation p2Luci reporters rather than the CCR5 -1 PRF signal as previously reported [36].

### 3.3. SFL Overexpression or Knockdown Results in 2-Fold Reciprocal Effects on -1 PRF

The ability, accuracy, and reproducibility of the bifluorescent reporter system was assessed by cloning validated PRF signals representing a range of frameshifting efficiencies and comparing the results with the same elements cloned into the dual luciferase reporter vector. The parallel analysis revealed that reporter systems yielded comparable PRF efficiencies, but that the bifluorescent system generated more reproducible data, particularly at higher PRF efficiencies (Appendix A). The bifluorescent reporter assay was used to monitor -1 frameshift efficiency in HEK293T cells over- or under-expressing SFL. Disruption of SFL homeostasis resulted in reciprocal two-fold changes to -1 PRF efficiencies in a panel of human and viral-derived -1 PRF signals (Figure 1). Consistent with previous studies, SFL overexpression resulted in ~2-fold decreases in -1 PRF efficiency promoted by the -1 PRF signals derived from SARS-CoV, HIV-1 and CCR5 (Figure 1). Specifically, SARS-CoV -1 PRF decreased from ~35% ± 5.5% in WT HEKs to ~18.9% ± 2.2% upon SFL overexpression (SFL OE) (*p* = 0.0057) (Figure 1A). SFL overexpression decreased HIV-1 frameshift efficiency from 4.5% ± 0.45% in WT to 2.89% ± 0.36% in (*p* = 0.157) (Figure 1B), and frameshifting promoted by the CCR5 -1 PRF signal decreased from 1.76% ± 0.13% in WT to 0.95% ± 0.21% (*p* = 0.0298) (Figure 1C). While significant decreases in -1 PRF due to SFL overexpression were observed using the bifluorescent reporter system, SFL overexpression did not yield significant decreases in -1 PRF using dual-luciferase reporters (Appendix A). In each individual experiment SFL overexpression did decrease -1 PRF as monitored by the dual luciferase reporters, but the variability in % -1 PRF between experimental replicates masked this effect.

Conversely, knockdown of expression using an shRNA targeting SFL (shSFL) resulted in a roughly two-fold increase in apparent frameshifting. SARS-CoV promoted -1 PRF increased from ~35% in WT HEKs to 51.2% ± 6.08% in SFL shRNA knockdown (shSFL) (*p* = 0.0081). HIV-1 stimulated -1 PRF increased from 4.5% in WT to 8.24% ± 1.65% in shSFL (*p* = 0.003), and CCR5 frameshifting increased from 1.76% in WT to 2.96% ± 0.51% (*p* = 0.0036). Similar results were observed using dual luciferase reporters (Appendix A).

### 3.4. SFL Is Not Limited to -1 PRF Signals

Although SFL was originally identified as a -1 PRF specific frameshift inhibitor [25], a recent study demonstrated SFL can also modulate programmed termination codon readthrough and that its activity is not limited to any specific type of translational recoding element [37]. To further probe the range of possible SFL substrates, SFL was over- or under-expressed in the context of cells expressing either a +1 PRF signal from ornithine decarboxylase antizyme 1 (OAZ1) [38] or a termination codon readthrough (TCR) signal from Venezuelan equine encephalitis virus (VEEV) [39,40,41]. Similar to the findings with -1 PRF signals, reciprocal two-fold changes were observed with these recoding signals. Specifically, SFL overexpression in OAZ1 decreased +1 PRF from 20.9% ± 1.85% to 12% ± 1.35% (*p* < 0.0001) and shSFL increased apparent frameshifting to 35.9% ± 1.78% (*p* < 0.0001), respectively (Figure 2A). SFL overexpression in VEEV decreased TCR from 1.25% ± 0.24% to 0.52% ± 0.05% (*p* = 0.0005) and shSFL increased TCR to 1.89% ± 0.22% (*p* = 0.0011) (Figure 2B).

To determine whether these effects are limited to *bona fide* recoding signals, the effects of SFL over- or under-expression were assayed using a spontaneous -1 frameshift (-1sFS) reporter in which the downstream (firefly luciferase) reporter is in the -1 frame relative to the upstream reporter. SFL overexpression did not significantly change -1 sFS, however SFL knockdown resulted in a 2.5-fold increase, from 0.66% ± 0.18% to 1.67% ± 0.57% (*p* = 0.0018) (Figure 2C). Following this observation, we examined the effects of SFL over- and under-expression on the 0-frame reporter, i.e., where both reporters are in-frame with one another. Although SFL overexpression did not significantly alter the 3′ to 5′ reporter ratio, knocking down SFL expression resulted in a 1.65-fold increase in reporter ratio (*p* < 0.0001) (Figure 2D). This suggests that in-frame mRNAs may also be substrates for SFL.

### 3.5. Disrupting SFL Homeostasis Reduces Reporter Protein Activity

The experiments described above examined ratios of ratios of two reporter proteins expressed in cells either over- or under-expressing SFL as compared to mock transfected cells. Since calculation of recoding efficiency can reduce information content by averaging ratios of ratios, an issue that particularly impacts dual luciferase-based assays due to their greater variability, measuring actual reporter gene activities presents a more accurate view of the effects of SFL on gene expression. Because SFL over- or under-expression influenced these ratios independent of the type, or even the presence of a recoding element, we examined actual reporter gene activities to determine the extent to which SFL directly influences reporter output. In WT HEK cells, expression of both AcGFP and mCherry from the bicistronic dual-fluorescence reporter was ~14,000 and ~1000 fluorescence units respectively per 150 uL of cell lysate. Over-expression of SFL promoted a ~2-fold decrease in both AcGFP and mCherry activities using this reporter (Figure 3A). Gene deletion of SFL (in isogenic SFL^-/-^ HEK cells) resulted in a 1.6-fold decrease in expression of the 5′ AcGFP reporter with no significant change in mCherry expression. Similar inhibitory trends of SFL over- or under-expression of gene expression were observed with 0-frame dual luciferase reporters (Figure 3B) and the -1 sFS dual luciferase reporter (Figure 3C). Over-expression of SFL promoted >10-fold decreases in both Renilla and firefly luciferase activities using the 0-frame reporter. SFL^-/-^ reduced reporter activities to a lesser extent, i.e., a ~6.8-fold in Renilla and ~5-fold in firefly (Figure 3B). We note that the baseline firefly luciferase activity is almost three orders of magnitude less than Renilla activity in the -1 sFS reporter construct, reflecting the baseline rate of spontaneous -1 frameshifting (Figure 3C). These observations suggested that SFL may normally monitor in-frame messages for spontaneous frameshifting activity. Consistent with this, the activity of a monocistronic GFP reporter was similarly impacted by SFL overexpression (>10-fold decrease) or in an isogenic SFL^-/-^ HEK cell line (~30% decrease) (Figure 3D).

### 3.6. Disruption of SFL Homeostasis Decreases mRNA Steady State Abundances

Given the negative impacts of increased or decreased SFL expression on reporter protein activities, we examined the effects on mRNA steady state abundances in isogenic WT HEK cells or HEK cells overexpressing SFL by qRT-PCR using 100 ng of total cDNA derived from cellular RNAs. SFL overexpression decreased the steady state abundances of mRNAs expressed from the GFP monocistronic reporter by 77% ± 5.2% *(p =* 0.0002), Renilla luciferase in the 0-frame dual luciferase reporter by 82% ± 13.9% *(p* = 0.0003), and Renilla luciferase in the SARS-CoV-2 -1 PRF reporter by 75.6% ± 26.1% (*p* = 0.0001) compared to WT HEKs. Importantly, SFL overexpression also decreased the abundance of the endogenous GAPDH mRNA by 44% ± 17.9% (*p* = 0.0021) (Figure 4).

In parallel, comparison of mRNA steady state abundances in isogenic WT HEK cells versus SFL^-/-^ HEK cells revealed that the lack of SFL also reduced the abundance of all four mRNAs, although to lesser extents. Specifically, in SFL^-/-^ cells, GFP mRNA was reduced by 25.6% ± 17.8% *(p =* 0.046), 0-frame Renilla mRNA levels by 53.7% ± 15% *(p =* 0.0001), SARS-CoV-2 Renilla mRNA levels by 72.4% ± 6% (*p* = 0.0002), and GAPDH levels were decreased by 40.9% + 14.3% (*p* = 0.0034) (Figure 4).

### 3.7. Genetic Analysis of SFL in the Context of Ribosome-Associated Quality Control (RQC)

RQC is initiated by disome formation when a trailing ribosome encounters a paused downstream ribosome [42,43]. SFL is known to bind tightly to hyper-rotated, frameshifted ribosomes, where it presumably causes them to pause [25,44]. This suggests that defects in SFL expression may affect the kinetics of pausing of frameshifted ribosomes, thus affecting RQC activation. Paused ribosomes direct recruitment of numerous factors including the GIGYF2-4EBP complex, ZNF598, and EDF1 [45]. GIGYF2-4EBP inhibits initiation by new ribosomes. Easily resolved pauses result in recruitment of the tRNA-eEF1A-GTP ternary complex, resulting in resumption of translation elongation. In contrast, persistent collisions stimulate eS10 and uS10 ubiquitination by ZNF598, which in turn stimulates ASCC3 recruitment to the disome [46,47,48]. This complex then recruits the no-go mRNA machinery which includes Pelota and Hbs1L.

The effects of SFL homeostasis disruption on RQC was tested by knocking down the expression of ZNF598, ASCC3, Hbs1L, or Pelota in HEK cells either overexpressing SFL, or in the SFL^-/-^ background using the in-frame dual luciferase reporter. shRNA knockdown of any of the RQC factors inhibited expression of the upstream Firefly reporter to approximately the same extent as SFL overexpression, but not as much as in SFL^-/-^ cells (Figure 5A). The combination of ZNF598 knockdown plus SFL overexpression was comparable to either of these conditions alone. In contrast, ZNF598 knockdown in SFL^-/-^ cells appeared to have a synergistic effect, reducing Renilla expression to an even greater extent than either of the two conditions alone. Similar patterns were observed upon shRNA knockdown of ASCC3 (Figure 5B), and Hbs1L (Figure 5C). In contrast, synergy between Pelota and SFL overexpression was not observed (Figure 5D). Rather, all conditions inhibited Renilla expression to similar extents. To probe potential positional effects, the effects of these conditions on expression of the in-frame downstream firefly luciferase reporter were also evaluated. These analyses revealed similar results, i.e., the absence of SFL had synergistic effects in combination with shRNA knockdown of ZNF598, ASCC3, and Hbs1L, while SFL overexpression was dominant to Pelota knockdown (Figure 5).

Nonsense-mediated mRNA decay (NMD) represents a second arm of the RQC apparatus. Importantly, ribosomes that encounter a termination codon in the wrong context have been observed to slip back and forth on mRNAs [49], suggesting that they too may recruit SFL. Knockdown of the NMD factor SMG1 also inhibited expression of the upstream Renilla reporter; there was no synergy with either SFL overexpression or lack of expression (Figure 5E). In contrast, expression of the downstream firefly reporter was more strongly affected by SFL overexpression, and lack of SMG1 in combination with SFL overexpression more closely resembled SMG1 knockdown than SFL overexpression alone, suggesting that SMG1 may be epistatic to SFL.

The effects of knocking down RQC factors in the context of the HIV-1 -1 PRF were also examined. Lack of ZNF598 enhanced expression of the downstream firefly reporter relative to upstream Renilla luciferase reporter, resulting in a net increase in -1 PRF efficiency (Appendix A). SFL overexpression resulted in a small but statistically insignificant decrease in the downstream reporter relative to the upstream one, resulting in a net decrease in -1 PRF efficiency (again, not statistically significant). The same patterns were observed upon shRNA knockdown of ASCC3, Hbs1L, Pelota, or SMG1 (Appendix A).

### 3.8. SFL Expression Is Significantly Reduced in Many Common Cancers and Correlates with Worse Clinical Outcomes

Given observations above, changes in SFL expression are expected to globally affect both the quality and quantity of protein synthesis. Although the transient alteration of SFL expression has therapeutically valuable antiviral effects, sustained defects in SFL expression should be pathological. To search for evidence of pathological effects, the Cancer Genome Atlas (https://www.cancer.gov/tcga (accessed on 28 February 2022)) and Genome Browser [34] databases were analyzed for changes in SFL expression across a panel of the most common cancer types [50] and their corresponding normal tissues. SFL expression was found to be decreased in all cancers examined by more than 30%, with the exception of cutaneous melanoma (SKCM) (Figure 6A, Appendix A). In addition, ZNF598 expression was also decreased in most cancer types except mesothelioma (MESO) (Figure 6B). Expression of other members of RQC including Pelota (Figure 6C), SMG1 (Figure 6D) did not correlate across cancer types, nor were patterns observed in the expression of reference genes (GAPDH, HNRPL, PCBP1, SNW1, and RER1) identified to have stable expression patterns in human cancer and matching normal cell types (Figure 6F–I) [51,52]. Strikingly, lower levels of SFL expression correlated with reduced survival in lung mesothelioma, bladder, and skin cancers (Appendix A).

## 4. Discussion

Over 30 years ago, multiple independent measurements of spontaneous ribosomal frameshifting generated the following paradox: long mRNAs encoding very large proteins (e.g., titin, alpha 5 laminin and BRCA2), are statistically untranslatable. While researching prior literature on SFL, we were struck by data indicating that this interferon-inducible antiviral protein appeared to be constitutively expressed, albeit at low levels, across a broad range of cell lines (see Figure 1D in [28]). Combining these two observations generated the hypothesis that constitutive low level SFL expression may enable the translational apparatus to identify and remove spontaneously frameshifted ribosomes while leaving mRNAs intact to be fully translated by trailing ribosomes. Corollary to this is the hypothesis that IFN induction of high levels of SFL may have evolved as an arm of the innate immune system to identify and disable viral mRNAs, many of which are known to program ribosomes to shift reading frame by >3 orders of magnitude.

The data shown in Figure 3 and Figure 4 are most informative: either too much or too little SFL results in decreased mRNA abundance (Figure 4), resulting in decreased protein expression (Figure 3). This held true regardless of the reporter used, or whether it encoded a synthetic bicistronic reporter or a natural monocistronic mRNA. This is a critical observation; it suggests that SFL functions to surveil cellular mRNAs for spontaneously frameshifted ribosomes. The ability of SFL to co-immunoprecipitate with Poly(A) binding protein cytoplasmic 1 (PABPC1p), La-motif related protein 1 (LARP1p), MOV10p, and UPF1p [28,53] further suggested a connection with the molecular machinery that is involved in translational surveillance in general. The reporter protein expression data shown in Figure 5 supports this, showing that disrupting SFL homeostasis in combination with knockdown of mRNAs encoding various elements of the RQC apparatus all promoted ~10-fold decreases in reporter protein activities. In this same figure, the ratios of downstream firefly to upstream Renilla luciferases generated from the in-frame reporter demonstrated that SFL overexpression tended to be dominant to shRNA knockdown of the RQC apparatus (Figure 5). In contrast, the effects of shRNA knockdown of SFL, which resulted in ~1.5-fold increases in reporter protein ratios, tended to remain the same or greater when combined shRNA knockdown of various RQC factors (Figure 5).

The synergistic effects of SFL knockdown shRNA knockdown of ZNF598, ASCC3, and Hbs1L (Figure 5) suggests SFL is recruited early in the process, stabilizing frameshifted ribosomes on mRNAs, thus enhancing disome formation, consistent with the early recruitment of these factors consequent to disome formation (reviewed in [54,55]). The dominance of SFL overexpression to Pelota knockdown also supports this view. We propose a model beginning with a frameshifted ribosome stalling on an mRNA (Figure 7A). Frameshifted ribosomes assume a hyper-rotated conformation [56], bringing the ribosomal proteins uL5 and eS31 in close enough proximity for SFL to interact with them and be recruited (see [44]). In this model, SFL recruits the release factors eRF1/3, terminating translation and initiating releasing the ribosome from the mRNA. Thus, SFL may function to save mRNAs from spontaneous frameshift events, therefore solving the problem of how long mRNAs are fully translated. Importantly, SFL concentration appears to matter. Current models suggest that the context and duration of ribosome collisions guide downstream consequences with increasing severity, ranging from rapid resolution of the problem which saves the mRNA for translation by successive ribosomes, all the way to complete translational shutdown and triggering of apoptotic pathways [57]. Accordingly, in the absence of SFL, frameshifted ribosomes are free to continue in the new reading frame where they will eventually either encounter a premature termination codon or continue translate into the 3′ UTR, in the former case activating NMD, and non-stop mRNA decay (NSD) in the latter (Figure 7B). This model is supported by the observation that the absence of SFL promoted a greater decrease on mRNA abundance when frameshifting rates are elevated over background by a -1 PRF signal (compare Figure 4A,B,D, with Figure 4C). It also follows that SFL constitutive under-expression would have globally negative impacts gene expression and cellular homeostasis, thus accounting for the link between low levels of SFL in many cancers (Figure 6) and the correlation between SFL expression and cancer survival rates (Appendix A). Conversely, overexpression (i.e., high concentrations) of SFL (Figure 7C) would stall frameshifted ribosomes for longer periods of time on the mRNA, providing more time for disome formation and recruitment of RQC factors, resulting in much larger decreases in mRNA abundance and gene expression. This is reflected by the data shown in Figure 4 and Figure 5. We suggest that this is the basis for the antiviral activity caused by SFL overexpression consequent to IFN induction (reviewed in [27]). By this model, IFN induction of SFL expression would represent an early step in the integrated stress response (reviewed in [58]) (Figure 7D). Elucidating when SFL is recruited during the RQC process, and the role played by ribosome collisions awaits deeper biophysical analyses.

Interestingly, SFL knockdown or knockout resulted in an ~2-fold increase in PRF (Figure 1 and Figure 2). The same observation was made when the space/time between elongating ribosomes was increased by decreasing translation initiation rates [59]. This two-fold effect was observed, regardless of the extent of initiation deficiency, and a model was suggested involving collisions between ribosomes stalled at a -1 PRF signal and the trailing ribosome [59]. Since then, the importance of ribosome collisions and RQC have been discovered (reviewed in [54,55,60,61]). We suggest that when ribosomes collide at a PRF signal, the leading one can either shift or not shift (the frequency of which is determined by the particular PRF signal), but the trailing ribosome cannot shift because the PRF stimulating mRNA structural element has been denatured by the transit of the leading ribosome and does not have enough time to reform before the sequence is covered by the lagging ribosome. In other words, only half of the ribosomes that would normally encounter a PRF signal can actually shift. This would explain the two-fold increase in PRF efficiency by either increasing the time between successive ribosomes or by the absence of SFL. Conversely, overexpression of SFL resulted in a two-fold decrease in PRF. How this works is complicated by the dynamics of mRNA decay, but points to the efficient clearance of frameshifted ribosomes. Illuminating the precise molecular mechanisms underlying how altering SFL expression results in the apparent two-fold changes in translational recoding promises to reveal novel insights into the biophysical interactions that occur between this class of cis-acting RNA elements and elongating ribosomes within the context of RQC.

We wish to address two final issues raised by the data presented here. First, the data presented in Figure 2C show that the rate of spontaneous ribosomal frameshifting is ~0.7 × 10^−3^, significantly larger than the 2 × 10^−4^ described in the prior literature. We suggest two possible explanations. One is that this is an inherent feature of bicistronic reporters. However, the activities of both the upstream and downstream reporter proteins in the spontaneous frameshift reporter are consistently an order of magnitude lower than those of the in-frame reporter, consistent with the former being a substrate for NMD (compare Figure 3B,C). Indeed, dual luciferase bicistronic reporters were first introduced into the translational recoding field to replace monocistronic reporters to control for apparent changes in recoding rates consequent to defects in NMD. This was because recoding reporters, which by their very nature are nonsense-containing mRNAs, are compared to in-frame control reporters (which are not NMD substrates) to generate recoding efficiency. Because of this discrepancy, the use of monocistronic reporters to measure recoding rates resulted in the mistaken conclusion that NMD factors influenced -1 PRF rates [62,63]. This raises the possibility that the 2 × 10^−4^ frequency of spontaneous frameshifting as originally measured using monocistronic reporters represents an underestimation of the frequency of these events. If substantiated, it would further emphasize the importance of SFL in surveilling mRNAs for spontaneous frameshifts. The second issue regards the data presented in Figure 1C and Appendix A showing that sequence derived from the human CCR5 mRNA promoted ~2% -1 PRF. A previous publication disputed the ability of this sequence to promote -1 PRF due to artifactual splicing issues with the first generation of dual luciferase reporters [36]. In the current work, the CCR5 sequence was cloned into the second generation dual luciferase reporter used by those authors (pSGDLuc) [20,36], the only difference being the insertion of *Bam* HI and *Sal* I restriction sites into the multiple cloning site of the vector (Appendix A). We also note that similar results were observed using the bifluorescent reporters (Figure 1C). RT PCR analyses suggest reporter splicing does indeed occur in the first-generation reporters but is not exclusive to the CCR5 sequence and can be seen in the 0-frame control reporter as well (Appendix A). The second-generation dual luciferase and bifluorescent reporters, however, lack such splicing artifacts regardless of frameshift sequence. Additionally, we did not observe the “reporter crash” seen in the prior study, consistent with absence of splicing artifacts in these reporter systems. Thus, at least in our hands, the CCR5-derived sequence appears to function as an efficient -1 PRF signal. The raw and collated data from these experiments are presented as Excel spreadsheets for the convenience of the reader (Appendix A). We leave it to the scientific community to resolve this matter.

## Figures and Tables

**Figure 1 viruses-15-02296-f001:**
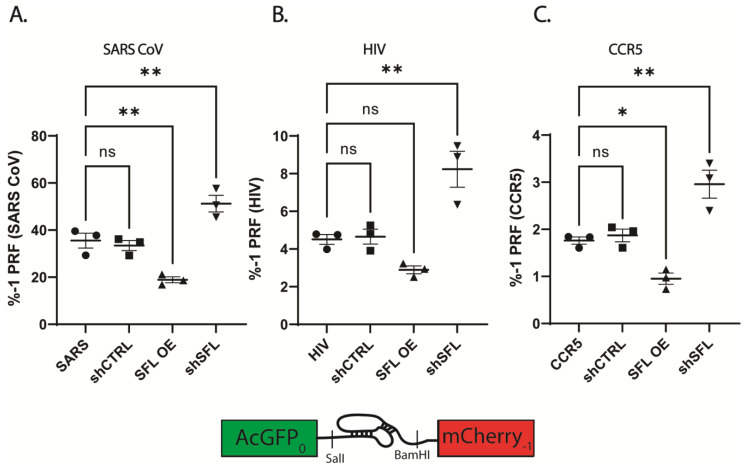
SFL overexpression or knockdown alters -1 PRF. Frameshift efficiency of three translational recoding elements measured using dual fluorescence reporters in HEK293T cells over or under-expressing SFL. (**A**) SARS-CoV -1 PRF signal, (**B**) HIV-1 -1 PRF signal, (**C**) CCR5 -1 PRF signal. Error bars denote standard error of the mean (SEM). * *p*-Value < 0.05, ** *p*-Value ≤ 0.01, ns, not significant.

**Figure 2 viruses-15-02296-f002:**
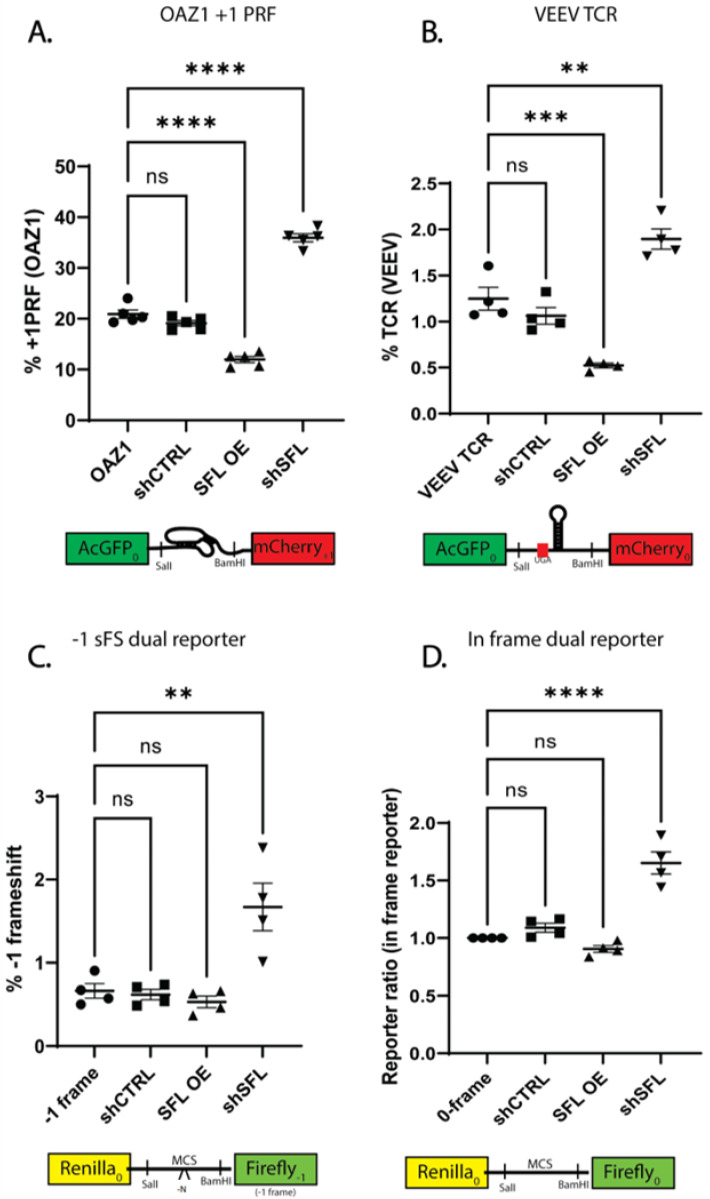
**SFL is not limited to -1 PRF signals.** Translational recoding efficiency and reporter ratio measured using dual fluorescence reporters in HEK293T cells over or under-expressing SFL. (**A**) OAZ1 +1 ribosomal frameshift signal, (**B**) VEEV termination codon readthrough signal, (**C**) -1 sFS dual luciferase reporter, (**D**) 0-frame dual luciferase reporter. Error bars denote SEM. ** *p*-Value ≤ 0.01, *** *p*-Value ≤ 0.001, **** *p*-Value ≤ 0.0001, ns not significant.

**Figure 3 viruses-15-02296-f003:**
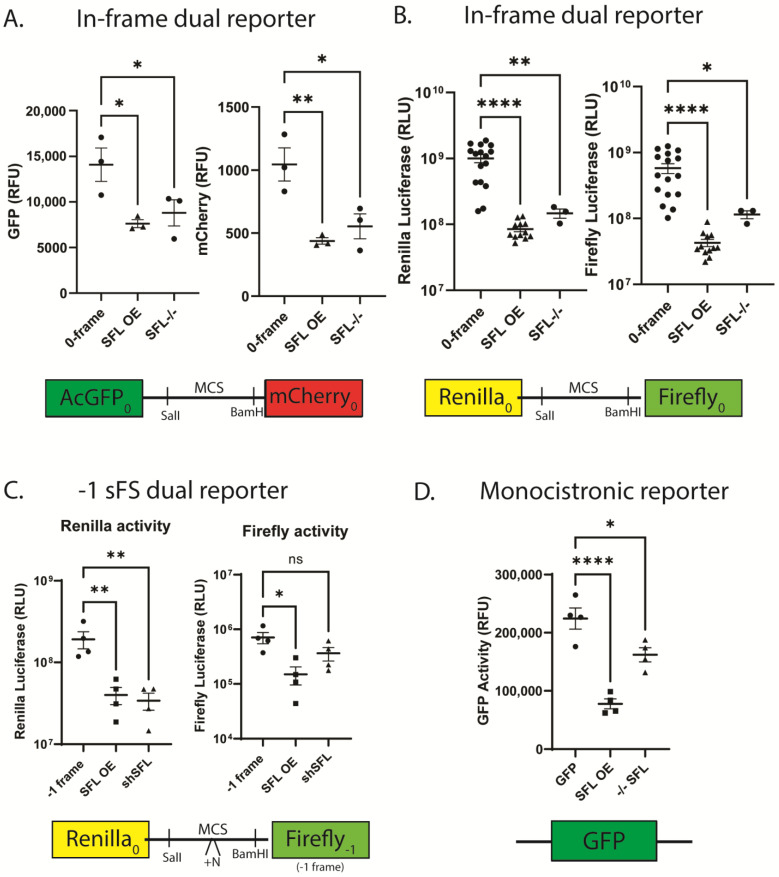
Disruption of SFL homeostasis inhibits reporter protein activity. Activity of reporters in HEK293T cells over or under-expressing SFL (**A**) 0-frame dual fluorescence reporter, (**B**) 0-frame dual luciferase reporter, (**C**) -1 sFS reporter, and (**D**) monocistronic GFP reporter. Error bars denote SEM. * *p*-Value < 0.05, ** *p*-Value ≤ 0.01, **** *p*-Value ≤ 0.0001, ns not significant.

**Figure 4 viruses-15-02296-f004:**
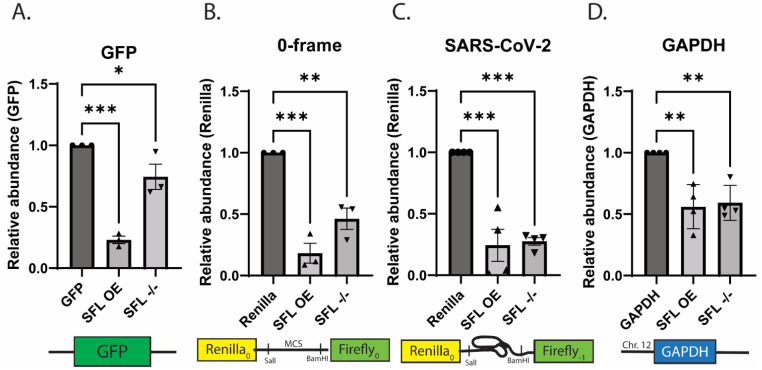
Disruption of SFL homeostasis decreases mRNA steady-state abundances relative to wild-type conditions. Relative mRNA abundance in SFL^+/+^ HEK293T or SFL^-/-^ HEK293T cells. Relative abundance of (**A**) GFP mRNA from a monocistronic reporter, Renilla luciferase mRNA from the (**B**) 0-frame control or (**C**) SARS-CoV-2 dual luciferase reporter, and (**D**) GAPDH mRNA. Error bars denote SEM. * *p*-Value < 0.05, ** *p*-Value ≤ 0.01, *** *p*-Value ≤ 0.001, ns not significant.

**Figure 5 viruses-15-02296-f005:**
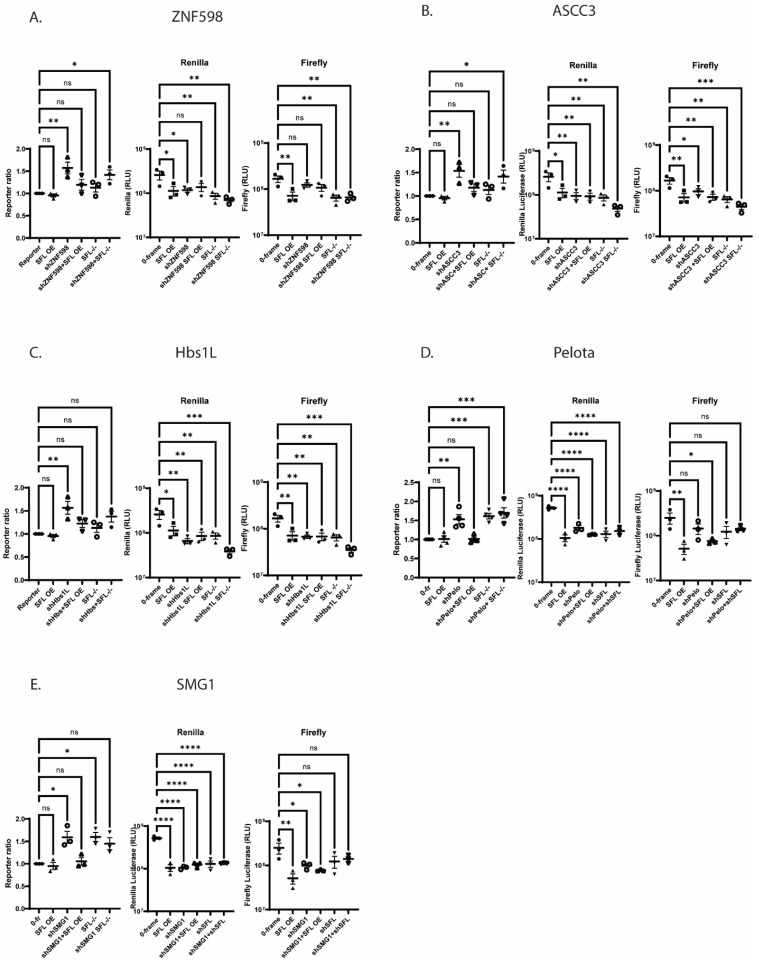
**SFL and RQC.** In-frame control reporter protein ratios and luciferase activity with shRNA knockdown of RQC proteins in WT HEK293T or SFL^-/-^ HEK293T. (**A**) ZNF598, (**B**) ASCC3, (**C**) Hbs1L, (**D**) Pelota, or (**E**) SMG1 shRNA knockdown. Error bars denote SEM. * *p*-Value < 0.05, ** *p*-Value ≤ 0.01, *** *p*-Value ≤ 0.001, **** *p*-Value ≤ 0.0001, ns not significant.

**Figure 6 viruses-15-02296-f006:**
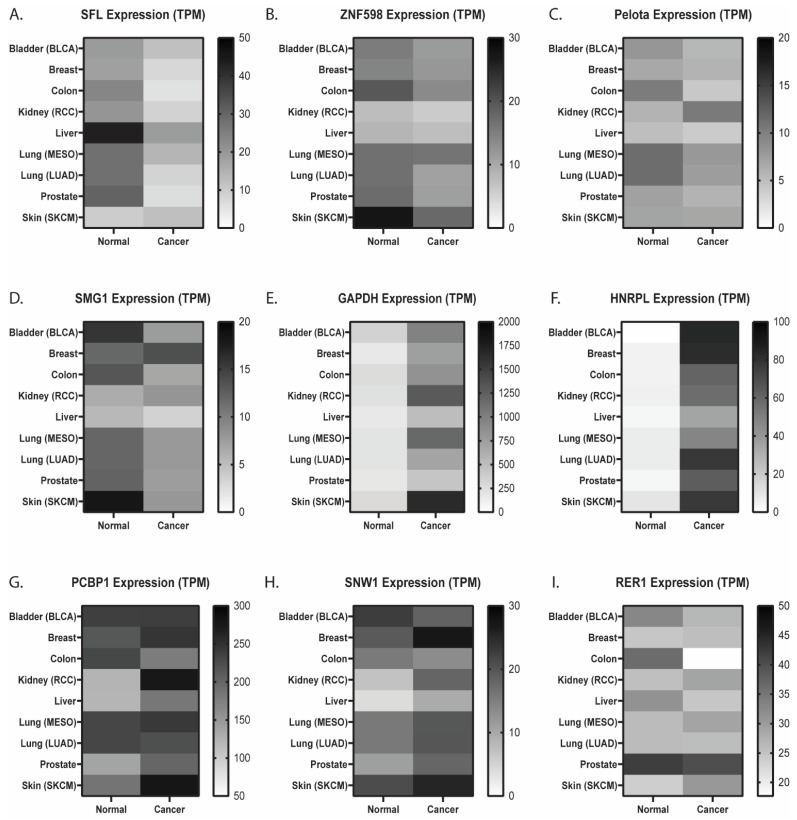
Gene expression pattern changes in cancer. Average TPM of RQC components or control genes in common cancers and corresponding normal tissue. (**A**) SFL, (**B**) ZNF598, (**C**) Pelota, (**D**) SMG1, (**E**) GAPDH, (**F**) HNRPL, (**G**) PCBP1, (**H**) SNW1, and (**I**) RER1. Shading intensity indicates the degree of SFL expression from high (dark) to low (light).

**Figure 7 viruses-15-02296-f007:**
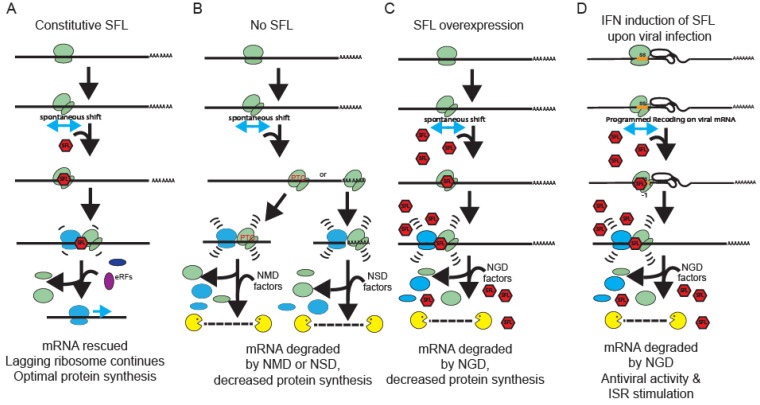
**Model.** (**A**) In constitutive low level SFL expression, SFL recognizes spontaneously frameshifted ribosomes, and locks them onto the mRNA. Disomes do not persist for long, as SFL recruits release factors and the frameshifted (leading) ribosome is removed from the mRNA. The lagging ribosome is free to continue translating the mRNA. (**B**) In the absence of SFL, frameshifted ribosomes are free to continue translating in the −1 or +1 frame. In most cases, they will encounter a PTC, activating mRNA degradation by NMD. In rare cases, they will continue translating past the in-frame termination codon and into the polyA tail, where they will activate NSD. (**C**) At high concentrations, SFL saturates the frameshifted ribosome, generating a long disome pause, activating NGD and decreasing protein expression. (**D**) Viral infection results in IFN induction of SFL expression, generating high concentrations of SFL. The resulting long pauses at -1 PRF signals results in disome stabilization, activating NGD and the ISR.

**Table 1 viruses-15-02296-t001:** shRNA target sequences used in *this work*.

shRNA Target	Target Sequence (5′ to 3′)	Cat#
SFL shA	GTGTATCCAACACGGATCCTC	TRCN0000434142
SFL shB	CCAAGAACTAAGTAACGATCT	TRCN0000161786
SFL shC	AGCAACCCTCACATTAGCAGT	TRCN0000420530
SFL shD	GAAGTTTCATGGGAAGGTATC	TRCN0000164654
SFL shE	GAAGTTCTGTGGGACACATTG	TRCN0000418723
ASCC3	TGAGGAGCGAACTGGATATTT	TRCN0000296023
Hbs1L	GCGATCTATTGACAAACCTTT	TRCN0000353597
PELO	GCAGTGAAGACCGACAACAAA	TRCN0000163394
SMG1	GCACTGTAACTACGGCTACAA	TRCN0000037413
ZNF598	CCAACCCTCTAAAGTTGGGAA	TRCN0000222610

**Table 2 viruses-15-02296-t002:** Plasmids used in this study.

Plasmid Name	Description
pJD2257	Modified pSGDluc (dual luciferase with inteins) with 0-frame control sequence insert
pJD2256	Modified pSGDluc (dual luciferase with inteins) with HIV-1 -1 PRF sequence insert
pJD2258	Modified pSGDluc (dual luciferase with inteins) with CCR5 -1 PRF sequence insert
pJD2359	Modified pSGDluc (dual luciferase with inteins) with SARS-CoV -1 PRF sequence insert
pJD2514	Modified pSGDluc (dual luciferase with inteins) with SARS-CoV-2 -1 PRF sequence insert
pJD2450	Modified pSGDluc (dual luciferase with inteins) with spontaneous -1 FS sequence insert
pJD2262	Dual fluorescent construct with 0-frame control sequence insert
pJD2261	Dual fluorescent construct with HIV-1 -1 PRF sequence insert
pJD2281	Dual fluorescent construct with CCR5 -1 PRF sequence insert
pJD2529	Dual fluorescent construct with SARS-CoV-2 sequence insert
pJD2350	Dual fluorescent construct with OAZ1 +1 PRF sequence insert
pJD2455	Dual fluorescent construct with VEEV stop codon readthrough sequence insert
pJD2260	pMAX GFP monocistronic reporter
pJD2612	C19orf66 ORF in pCMV-Myc
pJD175f	P2luci (first-generation dual luciferase) with 0-frame control sequence insert
pJD827	P2luci with CCR5 -1 PRF sequence insert

**Table 3 viruses-15-02296-t003:** Oligonucleotides.

Oligo Name	Sequence (5′ to 3′)
H_GAPDH_F	GGATGATGTTCTGGAGAGCC
H_GAPDH_R	CATCACCATCTTCCAGGAGC
GFP_qPCR_F	GGCTACGGCTTCTACCACTT
GFP_qPCR_R	CTCGTACTTCTCGATGCGGG
Firefly Luciferase qPCR Fwd	TCGCCTCTCTGATTAACGCC
Firefly Luciferase qPCR Rev	ATTACACCCGAGGGGGATGA
C19orf66_qPCR	Biorad PrimePCR qHsaCED0003572
C19orf66_RT_PCR_F	CTCAGGAAGGTGTGGAGCTG
C19orff_RT_PCR_R	GCCACTGCTAATGTGAGGGT
Pelo_qPCR_F	GACCGACAACAAACTGCTCCTG
Pelo_qPCR_R	AGCCACAGTAGGGTCACAAAGG
SMG1_qPCR_F	ATGCTGGTGAGCTTCGGCAGTA
SMG1_qPCR_R	CGCACATACACTTCAGGGTGGT
CRISPR validation primer Fwd	AAATCTGGCTTCTGAACCTCCT
CRISPR validation primer Rev	GTGGGAGACAAAGTGGACTGAG
SFL gRNA Top	CACCG GGATCCGTGTTGGATACACG
SFL gRNA Bottom	AAAC CGTGTATCCAACACGGATCC C
CRISPR validation primer Fwd	AAATCTGGCTTCTGAACCTCCT
CRISPR validation primer Rev	GTGGGAGACAAAGTGGACTGAG
-1 sFS_SDM_F	AAAGAGGCTGCGGCAAAAGC
-1 sFS_SDM_R	GCGGCTGCTTCGGTCGAC

## Data Availability

Data are available as Appendix A.

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
