# Peer review of "Shiftless Is a Novel Member of the Ribosome Stress Surveillance Machinery That Has Evolved to Play a Role in Innate Immunity and Cancer Surveillance"

_viruses, 2023, doi:10.3390/v15122296_

Round 1

Reviewer 1 Report

Comments and Suggestions for Authors

The manuscript by Kelly et al. demonstrated detailed analysis of the effects of overexpression and knock down/knock out of Shiftless on frame shift efficiency using reporter constructs and mRNA expression in reporter cells. The function of Shiftless and its potential role in translation surveillance are very important topic which have great interests to broad readers. This paper documented the effects of changing the expression level of shiftless on multiple aspects of protein translation from mRNA and is a good starting point towards the understanding of the function of Shiftless. The experiments appear to be designed reasonably and the data looks solid. With that being said, this reviewer also has several concerns/comments:

1. It is well documented in literature that Shiftless modulate the frameshifting efficiency of -1 PRF. Are there any studies on other recoding signals affected by Shiftless? If there are, please properly describe and cite these work in the paper. If this is the new data demonstrated by this work, this reviewer suggest highlight this in the discussion.

2. What are the effects of overexperssion or knock down/knock out of Shiftless on spontaneous frameshifting activity (not due to recoding signal)? The author put a lot of effort in describing changing homeostasis of Shiftless generally reduce the mRNA expression level but was not clear on the effects on spontaneous frameshifting activity. Make it clear if there are such effects. If not, the work is still good but may need to significantly revise the introduction section, which describes a lot of issues with transcribing long mRNA and thus gives the readers impression that Shiftless has a clear role in regulating this.

3. In general, the discuss section is a little too speculative.

4. Line 537, it appears Fig. 1C doesn't show the rate of spontaneous ribosomal framshifting.

Comments on the Quality of English Language

1. Line 31 "in order to"

2. Line 276, this is supposed to be Fig. S5?

Author Response

Reviewer #1:

The manuscript by Kelly et al. demonstrated detailed analysis of the effects of overexpression and knock down/knock out of Shiftless on frame shift efficiency using reporter constructs and mRNA expression in reporter cells. The function of Shiftless and its potential role in translation surveillance are very important topic which have great interests to broad readers. This paper documented the effects of changing the expression level of shiftless on multiple aspects of protein translation from mRNA and is a good starting point towards the understanding of the function of Shiftless. The experiments appear to be designed reasonably and the data looks solid. With that being said, this reviewer also has several concerns/comments:

  1. It is well documented in literature that Shiftless modulate the frameshifting efficiency of -1 PRF. Are there any studies on other recoding signals affected by Shiftless? If there are, please properly describe and cite these works in the paper. If this is the new data demonstrated by this work, this reviewer suggest highlight this in the discussion.

Response: The original submission noted that a prior paper (ref. 37, Napthine et al, Viruses, 2021) demonstrated that SFL also modulated stop codon readthrough. Thus, we chose to not make a big deal out of our confirmatory findings.

  1. What are the effects of overexpression or knock down/knock out of Shiftless on spontaneous frameshifting activity (not due to recoding signal)? The author put a lot of effort in describing changing homeostasis of Shiftless generally reduce the mRNA expression level but was not clear on the effects on spontaneous frameshifting activity. Make it clear if there are such effects. If not, the work is still good but may need to significantly revise the introduction section, which describes a lot of issues with transcribing long mRNA and thus gives the readers impression that Shiftless has a clear role in regulating this.

Response: Figure 2C, in which addition of a single base in the multiple cloning site, IS the spontaneous frameshift reporter. Only a spontaneous frameshift can generate the downstream reporter protein. Fig. 3C uses this same reporter to examine effects of SFL overexpression or shRNA knockdown on expression of the upstream (in frame) and downstream (out of frame) reporter. We have rephrased the text a little to (hopefully) make this more clear to readers

  1. In general, the discuss section is a little too speculative.

Response: We respectfully disagree.  The data are clearly telling us that SFL has a constitutive function, and our interpretation is wholly consistent with current thinking about the importance of ribosome collisions and mRNA decay.

  1. Line 537, it appears Fig. 1C doesn't show the rate of spontaneous ribosomal framshifting.

Response: We disagree. See response to item 2. Fixed, had Fig 1C written (CCR5), should be fig 2C (-1sFS).

Comments on the Quality of English Language

  1. Line 31 "in order to". Response: this has been fixed

  1. Line 276, this is supposed to be Fig. S5? Response: this has been fixed

Reviewer 2 Report

Comments and Suggestions for Authors

This quite interesting article shows how shiftless could be regulating the occurrence of frameshifting, with particular emphasis in its role on long mRNAs, encoding long proteins, where the ribosomes might stater and end up changing reading frame. To test this, they use bicistronic vectors with viral or cellular RNA regions that induce a necessary frameshifting. The authors propose that the correct levels or shiftless are necessary to maintain a correct avoidance of natural frameshifting, and that when shiftless is bound, will recruits proteins that will degrade the mRNA by two possible mechanisms. The data is quite clear and seems to me as that no other experiments are necessary, but there are several small things that must be improved before I can support publication of this article.

The logic behind point 3.2 is not clear to me, either then authors include a context that can allow the reader to understand why this was necessary, or simply eliminate it, because to me the article is supported without it.  

Figure S2A, should be in the main text in the corresponding section.

Surveyor assays are mentioned, but they should be included, I suggest they become figure S2A.

To make more amicable this article for people not familiar with frameshifting, a schematic of a generic constructs, indicating what happened when they are in frame, and when they are at -1 frame with the proteins to be detected, with be very useful, the mathematical analysis is explained in material and methods, but a visual explanation will be very useful.

For all figures with graphics the dataset is compared against the first bar which the basal or control condition, therefore all the brackets used to show the statistics are unnecessary. Significant statistic differences could be presents above each data set, indicating in the legend against whom they are compared, this will simplify a lot the figures.

All the legends should be more descriptive, for example the cartoon in figure 1 is not mentioned in the legend.

Legends must go immediately after the figure title. Right now, they look like another paragraph in the main text.

There are several inconsistencies in the text trough out the manuscript, for example 24-hours vs 24 hours, RT qPCR vs RT-qPCR, that must be check by the authors.

It is odd that some figures are not mentioned in order, for example Fig 2C is mentioned before fig 2B.

Each table should be presented immediately after the paragraph that mentioned it the first time, not all tables together.  

Table 3 indicated sequence 5´á 3´, it should be 5´-3´.

Lines 497-498 “data shown by the data”, what the authors mean?

The proposed model is quite nice, but the resolution of the figure needs to be improve.

Author Response

Reviewer #2

This quite interesting article shows how shiftless could be regulating the occurrence of frameshifting, with particular emphasis in its role on long mRNAs, encoding long proteins, where the ribosomes might stater and end up changing reading frame. To test this, they use bicistronic vectors with viral or cellular RNA regions that induce a necessary frameshifting. The authors propose that the correct levels or shiftless are necessary to maintain a correct avoidance of natural frameshifting, and that when shiftless is bound, will recruits proteins that will degrade the mRNA by two possible mechanisms. The data is quite clear and seems to me as that no other experiments are necessary, but there are several small things that must be improved before I can support publication of this article.

The logic behind point 3.2 is not clear to me, either then authors include a context that can allow the reader to understand why this was necessary, or simply eliminate it, because to me the article is supported without it.  

Figure S2A, should be in the main text in the corresponding section.

Surveyor assays are mentioned, but they should be included, I suggest they become figure S2A. Response: Surveyor assays were used as an initial screen for knockout cell lines before sequencing positive hits. As sequence data for the knockout cell line used is presented and clearly shows a base insertion into Shiftless, the authors believe it is not necessary to present Surveyor assay data.

To make more amicable this article for people not familiar with frameshifting, a schematic of a generic constructs, indicating what happened when they are in frame, and when they are at -1 frame with the proteins to be detected, with be very useful, the mathematical analysis is explained in material and methods, but a visual explanation will be very useful.

Response: Schematics of all reporter constructs are included with their respective figures throughout the manuscript and greater detail on mechanisms of frameshifting can be found in sources cited.

For all figures with graphics the dataset is compared against the first bar which the basal or control condition, therefore all the brackets used to show the statistics are unnecessary. Significant statistic differences could be presents above each data set, indicating in the legend against whom they are compared, this will simplify a lot the figures.

Response: The authors respectfully disagree. Brackets indicating which conditions are being compared in addition to significance indicators is standard and necessary.

All the legends should be more descriptive, for example the cartoon in figure 1 is not mentioned in the legend. Response: this has been fixed

Legends must go immediately after the figure title. Right now, they look like another paragraph in the main text. Response: this has been fixed

There are several inconsistencies in the text trough out the manuscript, for example 24-hours vs 24 hours, RT qPCR vs RT-qPCR, that must be check by the authors. Response: This is unclear. We don’t see these errors anywhere. For the 24-hour timeframe and 24 hour phrases are used but these are grammatically correct to the best of my knowledge

It is odd that some figures are not mentioned in order, for example Fig 2C is mentioned before fig 2B. Response: this was done for consistency between figures.  A minor issue.

Each table should be presented immediately after the paragraph that mentioned it the first time, not all tables together.  Response: this has been fixed

Table 3 indicated sequence 5´á 3´, it should be 5´-3´. Response: This is a dash in the original MS Word doc. Somehow it was changed to á in the conversion to a .pdf file.  Chalk this up the mysteries of Adobe.

Lines 497-498 “data shown by the data”, what the authors mean? Response: grammatical mistake corrected

The proposed model is quite nice, but the resolution of the figure needs to be improve. Response: The resolution is fine in the original. Again, this probably happened in conversion to a .pdf file.

Reviewer 3 Report

Comments and Suggestions for Authors

                This manuscript addresses the interesting issue of addressing spontaneous or programmed ribosome frameshifting during protein synthesis, including in the context of viral infections, via the protein Shiftless (which binds to shifting ribosomes and arrests translation.  The key observation made in the study is that changes in SFL levels result in activating mRNA quality control mechanisms and target mRNAs for degradation.

                Overall the data support the conclusions that are drawn and I believe that the study will be of significant interest to the translation/gene expression communities.  Two general comments.  First, the manuscript as currently presented is a rather dense read that may be challenging for the typical reader of Viruses to follow.  Thus I might suggest introducing questions and experiments in more general terms in the results section, as well as not relying as heavily on supplementary data.  Second, while I will leave this to the authors discretion, I was surprised to see a manuscript like this in a virus specialty journal when it largely focuses more on basic RNA biology.  However I leave that to the authors discretion.  I do have a few suggestions to polish the study:

Major Points:

1.       The efficiency of all KDs and the amount of all overexpressions should be shown to support various experiments throughout the Results section.

Minor Points

1.        The acronym SFL is introduced near the end of the abstract without being defined.

2.       3.1 and 3.2:  Maybe its just me, but it seems rather unconventional to have the first two subheadings/paragraphs of the results section rely on supplementary data.  I would recommend moving S2A and S3 (and perhaps even S4) into the main text. 

3.       Section 3.2:  I would recommend that the authors spend more time in the main text describing the CCR-1 PRF second generation reporters and putting the issue in better context to aid most general readers (particularly virologists) following/understanding the section.  The section is currently written for specialists with intimate knowledge of the constructs and the knowledge gap/issue being addressed in my opinion. 

4.       Fig. 6 legend.  Cancer should be lower case.

Author Response

Reviewer #3:

This manuscript addresses the interesting issue of addressing spontaneous or programmed ribosome frameshifting during protein synthesis, including in the context of viral infections, via the protein Shiftless (which binds to shifting ribosomes and arrests translation.  The key observation made in the study is that changes in SFL levels result in activating mRNA quality control mechanisms and target mRNAs for degradation.

                Overall the data support the conclusions that are drawn and I believe that the study will be of significant interest to the translation/gene expression communities.  Two general comments.  First, the manuscript as currently presented is a rather dense read that may be challenging for the typical reader of Viruses to follow.  Thus I might suggest introducing questions and experiments in more general terms in the results section, as well as not relying as heavily on supplementary data.  Second, while I will leave this to the authors discretion, I was surprised to see a manuscript like this in a virus specialty journal when it largely focuses more on basic RNA biology.  However I leave that to the authors discretion.  I do have a few suggestions to polish the study:

Major Points:

  1. The efficiency of all KDs and the amount of all overexpressions should be shown to support various experiments throughout the Results section.

Response:  We have added a few sentences in the results with overexpression and knockdown values from fig S2B

Minor Points

  1. The acronym SFL is introduced near the end of the abstract without being defined. Response: this had been fixed

  1. 3.1 and 3.2: Maybe its just me, but it seems rather unconventional to have the first two subheadings/paragraphs of the results section rely on supplementary data.  I would recommend moving S2A and S3 (and perhaps even S4) into the main text.

Response: We respectfully disagree.  Our reasoning is that Figs. S1 and S2 should be supplemental because, while necessary to the main story, Fig. S1 is not data per se, but is rather is simple stataistics, not real data.  Similarly, Fig. S2 is just “showing our work”, i.e. there to validate the systems that we used: these data are typically included as supplementary materials.  Lastly, although S3 could be used as a main figure since we spend a good amount of time talking about it in the discussion, it is not central enough to the main story that centers on Shiftless. Therefore, we chose to put it in the supplemental data.

  1. Section 3.2: I would recommend that the authors spend more time in the main text describing the CCR-1 PRF second generation reporters and putting the issue in better context to aid most general readers (particularly virologists) following/understanding the section.  The section is currently written for specialists with intimate knowledge of the constructs and the knowledge gap/issue being addressed in my opinion.

Response: We have expanded this section to include more context for why this was included.

  1. Fig. 6 legend. Cancer should be lower case. Response: This has been fixed.

Round 2

Reviewer 1 Report

Comments and Suggestions for Authors

Thanks for the responses. Before the manuscript could be accepted, the reviewer has just a few remaining questions to be addressed:

1. Please update the information for citation 36. The current date and title are incorrect.

2. Fig S3. For both the 0-frame cDNA and CCR5 cDNA data, pSGD has less DNA loaded into the gel compared with that of p2Luci. Are there internal controls to make sure the differences shown for p2Luci Vs pSGD are not due to a loading difference? It was claimed that "Small molecular weight products were observed in the first generation (p2luci) 0-frame control and CCR5 -1 PRF reporter mRNA samples... these PCR products were absent from the second-generation (pSGD) and bifluoresent (BiFl) reporter samples". It appears that if the amount of loaded DNA are similar, the p2Luci and pSGD bands will also be similar, ie. low molecular weight products are also present for the pSGD set of data.

3. Fig S3. "Additional PCR products of ~1200 and ~800 bp were observed with all the BiFl samples". What are these products? Could these be from off-target splicing products?

4. Fig 2. In the first two panels the author used BiFl reporter but in panels C and D used dual luciferase reporter. If "the bifluorescent system generated more reproducible data", why didn't use this BiFl reporter for all the experiments shown here?

5. This comment is related to my question 2 in the first round of review. Fig 2 panel C and D. If knocking down SFL increases the ratio of F-Luc to R-Luc for both the -1 sFS and in frame reporter, would this argue that the effect of SFL on the relative expressions of R- and F-Luc may not be due to the alteration of the spontaneous frameshifting (because expression of F-luc would need -1 sFS in the first reporter but not in the second reporter)? If this is true, the function of SFL in spontaneous frameshiting is still unknown and the efforts trying to connect long mRNA translation issue and function of SFL would still lack evidence.

6. Did the author define how the % PRF is calculated? If not, please add this information, which will make it easier for the reader to understand this work rather than having to refer to previous publications.

Author Response

Responses to Reviewer 1

  1. Please update the information for citation 36. The current date and title are incorrect.

Response: This has been fixed

  1. Fig S3. For both the 0-frame cDNA and CCR5 cDNA data, pSGD has less DNA loaded into the gel compared with that of p2Luci. Are there internal controls to make sure the differences shown for p2Luci Vs pSGD are not due to a loading difference? It was claimed that "Small molecular weight products were observed in the first generation (p2luci) 0-frame control and CCR5 -1 PRF reporter mRNA samples... these PCR products were absent from the second-generation (pSGD) and bifluoresent (BiFl) reporter samples". It appears that if the amount of loaded DNA are similar, the p2Luci and pSGD bands will also be similar, ie. low molecular weight products are also present for the pSGD set of data.

Response: We have added a second, independent replicate of this experiment to the Supplemental data.  The reason why pSGD appears to have a splicing problem, it’s not our reporter…the reviewer is advised to take this up with its creators.  As noted in the original Cover Letter, this is last research paper from the Dinman lab.  It was performed on a budgetary shoestring, and we did not have the funding to chase down artifacts.

  1. Fig S3. "Additional PCR products of ~1200 and ~800 bp were observed with all the BiFl samples". What are these products? Could these be from off-target splicing products?

Response: again, we did not have the funds to waste on chasing down artifacts.

  1. Fig 2. In the first two panels the author used BiFl reporter but in panels C and D used dual luciferase reporter. If "the bifluorescent system generated more reproducible data", why didn't use this BiFl reporter for all the experiments shown here?

Response: Typically, when one is introducing new technology, comparisons are made to the old one. That was the case here.

  1. This comment is related to my question 2 in the first round of review. Fig 2 panel C and D. If knocking down SFL increases the ratio of F-Luc to R-Luc for both the -1 sFS and in frame reporter, would this argue that the effect of SFL on the relative expressions of R- and F-Luc may not be due to the alteration of the spontaneous frameshifting (because expression of F-luc would need -1 sFS in the first reporter but not in the second reporter)? If this is true, the function of SFL in spontaneous frameshiting is still unknown and the efforts trying to connect long mRNA translation issue and function of SFL would still lack evidence.

Response:  Interesting interpretation.  Our interpretation is that, in the absence of SFL, relatively more ribosomes are getting through.  Please keep in mind that in this analysis, we are measuring ratios of ratios. The really critical data are in Figs. 3C,D and 4B. These show the direct effects on reporter activity  and mRNA abundances.  The reviewer is overanalyzing this.  

  1. Did the author define how the % PRF is calculated? If not, please add this information, which will make it easier for the reader to understand this work rather than having to refer to previous publications.

Response: % PRF calculations are explained in the text in section 2.6 for luciferase reporters and in 2.7 for bifluorescent reporters by referencing Munshi et al (currently ref 33).